# Integrated Transcriptomic and Metabolomic Analyses Reveal Low-Temperature Tolerance Mechanism in Giant Freshwater Prawn *Macrobrachium rosenbergii*

**DOI:** 10.3390/ani13101605

**Published:** 2023-05-11

**Authors:** Haihui Tu, Xin Peng, Xinyi Yao, Qiongying Tang, Zhenglong Xia, Jingfen Li, Guoliang Yang, Shaokui Yi

**Affiliations:** 1Zhejiang Provincial Key Laboratory of Aquatic Resources Conservation and Development, Key Laboratory of Aquatic Animal Genetic Breeding and Nutrition, Chinese Academy of Fishery Sciences, College of Life Sciences, Huzhou University, Huzhou 313000, China; tuhaihui09@163.com (H.T.); pengxin0521@163.com (X.P.); yaoxinyi2002@163.com (X.Y.); lijingfen@zjhu.edu.cn (J.L.); ygl0572@163.com (G.Y.); 2Jiangsu Shufeng Prawn Breeding Co., Ltd., Gaoyou 225654, China; zjhill@126.com

**Keywords:** low-temperature tolerance, gene regulation, metabolite, lipid metabolism, energy metabolism

## Abstract

**Simple Summary:**

*Macrobrachium rosenbergii* is a kind of warm-water species, and water temperature can affect its growth, metabolism, and even survival. We investigated the gene regulation patterns and identified the differential metabolites between the low-temperature tolerant and sensitive groups using RNA-seq and metabolomic methods. We found that the genes and metabolites associated with lipid metabolism and energy metabolism play vital roles in responding to low-temperature stress. This study provides a molecular basis for the selection of a low-temperature tolerant strain of the freshwater prawn, *Macrobrachium rosenbergii*.

**Abstract:**

Water temperature, as an important environmental factor, affects the growth and metabolism of aquatic animals and even their survival. The giant freshwater prawn (GFP) *Macrobrachium rosenbergii* is a kind of warm-water species, and its survival temperature ranges from 18 °C to 34 °C. In this study, we performed transcriptomic and metabolomic analyses to clarify the potential molecular mechanism of responding to low-temperature stress in adult GFP. The treatments with low-temperature stress showed that the lowest lethal temperature of the GFP was 12.3 °C. KEGG enrichment analyses revealed that the differentially expressed genes and metabolites were both enriched in lipid and energy metabolism pathways. Some key genes, such as *phosphoenolpyruvate carboxykinase* and *fatty acid synthase*, as well as the content of the metabolites dodecanoic acid and alpha-linolenic acid, were altered under low-temperature stress. Importantly, the levels of unsaturated fatty acids were decreased in LS (low-temperature sensitive group) vs. Con (control group). In LT (low-temperature tolerant group) vs. Con, the genes related to fatty acid synthesis and degradation were upregulated to cope with low-temperature stress. It suggested that the genes and metabolites associated with lipid metabolism and energy metabolism play vital roles in responding to low-temperature stress. This study provided a molecular basis for the selection of a low-temperature tolerant strain.

## 1. Introduction

The giant freshwater prawn (*Macrobrachium rosenbergii*; GFP) has been widely farmed worldwide due to its wide feeding, fast growth, short breeding cycle, and high economic value [1]. In China, the GFP production in 2021 reached 171,263 tons, accounting for more than half of the global production [2]. As a warm-water aquatic species, it can tolerant temperatures ranging from 14 °C to 35 °C, and the optimum temperature for nursery rearing is limited to 26~31 °C [3]. A recent study showed that the lowest lethal temperature of juvenile GFP is about 11.5 °C [4]. However, in most areas of China, the water temperature is much lower than the optimum survival temperature for the majority of the year. In GFP aquaculture practices, the heating of rearing water entails a huge energy consumption due to the use of a boiler in winter and early spring.

Water temperature is an important environmental factor in aquatic animals, and previous studies have shown that low-temperature stress could induce oxidative stress [5], change osmolality [6] and cell membrane fluidity [7], reduce muscle fiber density [8] and mitochondrial density [9], and lead to cell apoptosis [10]. At the molecular level, several significantly differentially expressed genes related to lipid metabolism, anti-oxidant stress, signal transduction, and immune response [11,12,13] were identified. For example, *acyl-CoA oxidase 1* (*acox1*), which was related to the biosynthesis of unsaturated fatty acid, was significantly upregulated in longsnout catfish (*Leiocassis longirostris*) under low-temperature stress [14]. The expression levels of the *heat shock protein* (*HSP*) gene family including *HSP10*, *HSP70*, and *HSP90* were significantly increased under low-temperature stress, indicating that these genes are vital to low-temperature tolerance in GFP [4]. Moreover, the expression levels of *HSP70* and *HSP90* in Antarctic notothenioid fish were higher than those in normal-temperature fish, indicating that *HSPs* were essential to assist protein folding under low temperatures [10]. However, the mechanisms of adaptive regulation under low-temperature stress are still unknown.

Transcriptomic and metabolomic analyses have been widely used in the studies of aquatic animals, for example, in *Marsupenaeus japonicus*, the integration of transcriptomic and metabolomic data showed that the pathways related to energy metabolism were down-regulated while the production of ATP and unsaturated fatty acid were up-regulated with low-temperature stress [15]. On the other hand, the differentially expressed genes and metabolites of *Cyprinus carpio* exposed to silver nanoparticles are mainly involved in the metabolism of amino acids, carbohydrates, and lipids, as well as the tricarboxylic acid cycle (TCA) cycle [16].

In this study, we performed transcriptomic and metabolomic analyses of hepatopancreas to reveal the potential molecular mechanisms of low-temperature tolerance in adult GFP. Firstly, the GFP samples were separated and selected via the treatment of extremely low temperature, and three groups, including the low-temperature tolerant group (LT), the low-temperature sensitive group (LS), and the control group (Con), were used for the transcriptomic and metabolomic analyses. The primary objective of this study was to find the key genes and metabolites that participate in thermal adaption of GFP, and the molecular regulation mechanism of low-temperature tolerance in GFP. This study provides a basis for the selective breeding of low-temperature tolerant GFP strains.

## 2. Materials and Methods

### 2.1. Low-Temperature Stress Treatment and Sampling

A total of 1320 adult prawns (about 200 days old, weight 47.73 ± 12.73 g, body length 123.60 ± 16.90 mm) from 22 families were obtained from Jiangsu Shufeng Prawn Breeding Co., Ltd. (Gaoyou, China). All prawns were temporarily cultured in concrete ponds with a density of about 12~14 individuals/m^2^ under the same condition.

The low-temperature stress experiment started after one week of temporary culture. The cooling rate was 0.25 °C/h by adding ice cubes, and prawn mortality was observed hourly during the cooling period. A prawn was identified as dead when its body tilted to one side, has become unresponsive to stimuli from the external environment, and with no gill movement. The temperature at death (TAD) and the time at death were recorded, and the phenotypic traits including body weight (BW) and body length (BL) were measured until all prawns died. The first and last 5% of mortalities were defined as the low-temperature sensitive group (LS, limit temperature 13 °C) and the low-temperature tolerant group (LT, limit temperature 12.3 °C), respectively. Six individuals in each of the LS, LT, and Con groups (25 °C, no death) were dissected for hepatopancreas collection. All samples were snap-frozen in liquid nitrogen and stored at −80 °C for transcriptomic and metabolomic analysis.

### 2.2. Transcriptomic Analysis

#### 2.2.1. Total RNA Extraction, Library Preparation, and Sequencing

Three individuals for each of the three groups (i.e., LS, LT, Con) were used for RNA sequencing. Total RNA extraction, quality inspection, library construction, and the following Illumina sequencing were performed by Metware Biotechnology Co., Ltd. (Wuhan, China).

#### 2.2.2. Transcript Assembly and Annotation

Transcriptome assembly was performed from clean reads using Trinity (v2.11.0) [17], and regroup relevant transcripts into “gene” clusters using Corset [18]. Diamond [19] or HMMER [20] were used to annotate gene function through the following seven databases: Nr (NCBI non-redundant protein sequences), Swiss-Prot (a manually annotated and reviewed protein sequence database), Trembl (a variety of new documentation files and the creation of TrEMBL, a computer annotated supplement to Swiss-Prot), KEGG (Kyoto Encyclopedia of Genes and Genomes), GO (Gene Ontology), KOG (euKaryotic Ortholog Groups), and Pfam (Protein family).

#### 2.2.3. Differentially Expressed Gene Analysis

Gene expression levels were estimated via RSEM [21], and then the Fragments Per Kilobase of transcript per Million mapped reads (FPKM) of each gene were calculated. The differential expression between the two groups was analyzed using DESeq2 v1.22.1 [22,23]. The *p*-value was corrected using the Benjamini–Hochberg method. All unigenes with |log_2_foldchange| ≥ 1 and *p*-value (*P*adj) < 0.05 were labeled as differentially expressed genes (DEGs). The GO and KEGG enrichment analysis was performed based on the hypergeometric test.

### 2.3. Metabolomic Analysis

#### 2.3.1. Extraction of Metabolites and LC-MS Analysis

Six individuals for each of the three groups (i.e., LS, LT, Con) were used for metabolome analysis. Metware Biotechnology Co., Ltd. (Wuhan, China) extracted the metabolites from each prawn’s hepatopancreas for metabolome analysis using UPLC (1290 Infinity LC, Agilent Technologies, Santa Clara, CA, USA) and MS (QTOF/MS-6545, Agilent Technologies, Santa Clara, CA, USA).

#### 2.3.2. Metabolome Data Processing

The raw data generated by LC-MS were converted to mzML format using the software ProteoWizard v3.0 [24], and the XCMS program [25] was used for peak extraction, alignment, and retention time correction. After that, metabolic identification information was obtained by searching the laboratory’s self-built database [26], integrated public database (including Metlin, HMDB, KEGG, Mona, and MassBank), AI database [27], and MetDNA [28].

For the multivariate statistical analysis, principal component analysis (PCA) and orthogonal projections to latent structures-discriminant analysis (OPLS-DA) were performed using the R platform. The Variable Important for the Projection (VIP) values were extracted from the OPLS-DA result and calculated to reveal the metabolite expression patterns. Significantly differential metabolites (DMs) between groups were determined by VIP value (VIP ≥ 1), *p*-value (*p*-value < 0.05), and absolute log_2_foldchange (|log_2_foldchange| ≥ 1). Metabolites were annotated and then mapped to the KEGG pathway database. For a given list of metabolites, significantly enriched pathways were identified by *p*-value from the hypergeometric test.

### 2.4. Combination of Transcriptomic and Metabolomic Analyses

For their pairwise comparisons including LS vs. Con, LT vs. Con, and LT vs. LS, all DEGs and DMs were mapped to the KEGG pathway database to explore the relationship between genes and metabolites. The changed pathways were confirmed by the KEGG pathway enrichment analysis. Interaction networks related to differentially expressed genes and metabolites were plotted by the Cytoscape v3.8 software [29] to reveal the relationship between DEGs and DMs under low-temperature stress.

### 2.5. Validation of Selected Genes by RT-qPCR

Ten genes (Appendix A) were selected to validate the reliability of transcriptome results using real-time quantitative PCR (RT-qPCR). The RT-qPCR was performed with TB Green^®^ Premix Ex TaqTM II (Takara, Dalian, China) using CFX96TM Real-Time PCR System (Bio-Rad, Hercules, CA, USA). The expression level of target genes was calculated by the 2^−∆∆^CT method [30].

### 2.6. Statistics Analysis

The cooling degree hours (CDH) were calculated using the following formula [31,32]:$$CDH=\sum_{i=1}^{k}\left[t_{i}\times (T_{0}-T_{i})\right]$$
where *t_i_* is the difference in cooling temperature from *i* + 1 to *i*, *T*_0_ is the temperature of the first dead prawn of GFP adults at low-temperature stress, *T_i_* is the temperature at moment *i*, and *k* is the time point of detection recorded when the test prawn died.

Data of phenotypic traits (BL, BW, CDH, TAD) and RT-qPCR were expressed as the mean ± standard deviation (SD).

## 3. Results

### 3.1. Evaluation of Low-Temperature Tolerance

Prawns were stressed with the gradual decline in water temperature from 22 °C to 12 °C. When the temperature decreased to 14.2 °C, prawns began to die and this temperature was performed as the primary temperature to calculate CDH. With the temperature decreasing, the cumulative mortality reached 41.25% at 12.5 °C and CDH was 141.12 °C·h. All the prawns died when the temperature declined to 12.3 °C and CDH increased to 171.28 °C·h (Figure 1, Table 1).

### 3.2. Identification, GO, and KEGG Classification of DEGs under Low-Temperature Stress

RNA-seq generated 41,858,116~49,724,804 clean reads from nine libraries. The average value of Q20 and Q30 were 97.63% and 93.64%, respectively (Appendix A). A total of 163,845 transcripts with an average length of 664 bp were assembled. Furthermore, 144,882 unigenes were obtained with an average length of 719 bp and N50 of 1501 bp (Appendix A). Among all unigenes, 35,220 (24.31%) were annotated in at least one database (Appendix A).

All unigenes with |log_2_foldchange| ≥ 1 and *p*adj < 0.05 were identified as DEGs from three comparative groups. The total number of DEGs in LS vs. Con was 1073, including 236 up-regulated and 837 down-regulated. With the decline of temperature, the number of DEGs in LT vs. Con were rapidly increased to 10,999, and the number of up-regulated DEGs (8489) was obviously more than those of down-regulated DEGs (2510), whereas the DEGs were few in LT vs. LS with only 498 up-regulated and 264 down-regulated DEGs (Figure 2A). The Venn plot showed that there were only 3 of the same DEGs in 3 groups, and a total of 1461 overlapped DEGs in 3 comparison groups (Figure 2B). The heatmap of 1461 overlapped DEGs is shown in Figure 2C, which indicates that both LS and LT were different from Con.

DEGs in three comparative groups were classified into three GO functional categories: biological process (BP), cellular component (CC), and molecular function (MF) (Appendix A). In BP, most DEGs in every comparative group were enriched in “Cellular process”, “Metabolic process” and “Biological regulation”. In CC, “Cell” was the most commonly represented, followed by “Cell part” and “Organelle”. The top three clusters in MF were “Binding”, “Catalytic activity”, and “Transporter activity”.

The top 20 KEGG pathways enriched in each comparative group are shown in Figure 3. In LS vs. Con, DEGs were significantly enriched in “Glycerophospholipid metabolism”, “Lysine degradation”, “Endocytosis” and “Spliceosome”. With the temperature dropped, significant enrichment pathways of DEGs increased in LT vs. Con, and the top five pathways were “Endocytosis”, “Spliceosome”, “Wnt signaling pathway”, “SNARE interactions in vesicular transport” and “Lysine degradation”, while only three pathways were significantly enriched in LT vs. LS, including “Oxidative phosphorylation”, “ECM-receptor interaction” and “Mucin type O-glycan biosynthesis”. The same significant pathways enriched in two comparison groups with Con were “Glycerophospholipid metabolism”, “Lysine degradation”, “Endocytosis”, and “Spliceosome”.

### 3.3. DEGs Related to Lipid and Energy Metabolism

Enriched pathways of DEGs in lipid and energy metabolism are shown in Table 2. In comparison groups with Con, including LT vs. Con and LS vs. Con, DEGs were enriched in “Glycerophospholipid metabolism”, “Glycerolipid metabolism”, “Biosynthesis of unsaturated fatty acids”, “Fatty acid elongation”, and “Glycolysis/Gluconeogenesis”, “Oxidative phosphorylation”, “Citrate cycle (TCA cycle)”. In LT vs. Con, DEGs were also enriched in “Fatty acid biosynthesis” and “Fatty acid degradation”. However, only “Glycerophospholipid metabolism”, “Fatty acid metabolism”, “Fatty acid degradation”, and “Oxidative phosphorylation” were enriched in LT vs. LS.

Potential genes related to low-temperature tolerance were selected in related pathways, and the expression changes are shown in Appendix A. Compared with the control group, the experiment group including LT vs. Con and LS vs. Con under low-temperature stress showed that the expression of four genes including *1-acyl-sn-glycerol-3-phosphate acyltransferase* (*AGPAT*), *17beta-estradiol 17-dehydrogenase* (*HSD17B12*), and *phosphoenolpyruvate carboxykinase* (*PCK*) were increased while *triacylglycerol lipase* (*TGL*) was downregulated. In LT vs. Con, some DEGs such as *fatty acid synthase* (*FAS*) and *citrate synthase* (*CS*) were downregulated, but more DEGs such as *3-oxoacyl-*[*acyl-carrier-protein*] *synthase II* (*FabF*), *malonyl-CoA-acyl carrier protein transacylase* (*FabD*), *elongase of very long chain fatty acids* (*ELOVL*), *acyl-coenzyme A thioesterase* (*ACOT*), *succinate dehydrogenase* (*SDH*) and *lactate dehydrogenase* (*LDH*) were upregulated. The expression of *acyl-CoA dehydrogenase* (*ACADM*) increased in LT vs. LS.

### 3.4. Dynamics of Metabolites under Low-Temperature Stress

To investigate the metabolic changes in the hepatopancreas of GFP under low-temperature stress, untargeted metabolomic analyses were performed using UPLC–MS platform. The repeatability of metabolite extraction and detection can be determined by Pearson correlation analysis, variation analysis, and PCA (Appendix A). The PCA plots of three comparative groups suggested that metabolites had significantly changed under low-temperature stress (LT vs. Con and LS vs. Con), while the difference between LT and LS was little (Appendix A).

There were 47 up-regulated and 112 down-regulated DMs in LS vs. Con. The number of DMs increased with the temperature dropped (Figure 4A). The top 20 DMs of LS vs. Con and LT vs. Con in positive or negative ion mode are shown in Figure 4B,C, respectively.

To further identify the metabolic pathway potentially affected by low-temperature stress, identified DMs were enriched using KEGG pathway analysis and the result showed that DMs varied widely among three comparison groups (Appendix A). “ABC transporters”, “Biosynthesis of unsaturated fatty acids”, “Fatty acid biosynthesis” and “Citrate cycle (TCA cycle)” were significantly enriched in LS vs. Con. With the temperature decline, the pathways changed. In LT vs. Con, the enriched pathways were “cAMP signaling pathway”, “ABC transporters”, “Propanoate metabolism” and “Metabolic pathways”. The enriched pathways of DMs between LT and LS were much less than the other two comparison groups, only including “Arachidonic acid metabolism”, “Tyrosine metabolism”, “cAMP signaling pathway” and “Glycerophospholipid metabolism”.

### 3.5. Differential Metabolites Related to Lipid and Energy Metabolism

In metabolomics, enriched pathways of DMs in lipid and energy metabolism were shown in Table 3. DMs between the experiment group and control group were enriched in “Fatty acid biosynthesis”, “alpha-Linolenic acid metabolism”, “Citrate cycle (TCA cycle)”, and “Oxidative phosphorylation”. “Biosynthesis of unsaturated fatty acids” and “Arachidonic acid metabolism” were also enriched in LS vs. Con. There were three related pathways enriched in LS vs. LT, including “Regulation of lipolysis in adipocytes”, “Glycerophospholipid metabolism” and “Arachidonic acid metabolism”.

DMs in lipid and energy metabolism pathways were shown in Appendix A. Most of the DMs decreased in LS vs. Con, such as dodecanoic acid (FFA(12:0)), alpha-linolenic acid (ALA) (FFA(18:3)), linolenic acid (LA) (FFA(18:2)), icosadienoic acid (FFA(20:2)), and succinic acid, while stearic acid (FFA(18:0)) upregulated. In LT vs. Con, dodecanoic acid (FFA(12:0)) and succinic acid are down-regulated, whereas traumatic acid is up-regulated. 5-Oxo-ETE, L-noradrenaline, and triethanolamine increased in LS vs. LT.

### 3.6. Correlation Analysis of Transcriptomic and Metabolomic Data

The relationship between genes and metabolites was further revealed using combined transcriptomic and metabolomic analyses. Twenty-two common pathways were enriched in LS vs. Con and LT vs. Con, including “ABC transporters”, “Citrate cycle (TCA cycle)”, “Oxidative phosphorylation”, and “Pyruvate metabolism”. “Biosynthesis of unsaturated fatty acids” was enriched in LS vs. Con, and “Fatty acid biosynthesis” in LT vs. Con. In LT vs. LS, DEGs and DMs were enriched in “Glycerophospholipid metabolism” and “Metabolic pathways”.

Lipid metabolism pathways were important for coping with low-temperature stress. The potential regulatory network figure showed that acetyl-CoA was in the central position and was associated with both lipid and energy metabolism (Figure 5). Here, fatty acid metabolism was essential in response to low-temperature stress, and the interaction networks of DEGs and DMs were plotted with an absolute value of the correlation coefficient greater than 0.8 in “Biosynthesis of unsaturated fatty acids” of LS vs. Con (Figure 6A) and “Fatty acid biosynthesis” of LT vs. Con (Figure 6B). The results showed that LA, icosadienoic acid, and stearic acid were closely related to *HSD17B12*, and only stearic acid was positively correlated with *HSD17B12*, there is a negative correlation between *ACSL* and dodecanoic acid, *MECR*, and dodecanoic acid.

### 3.7. Validation of Significant DEGs by qRT-PCR

Ten DEGs were selected to verify the reliability of RNA-seq using RT-qPCR (Appendix A). In RT-qPCR, two DEGs including *FAS* and *acetyl-CoA carboxylase* (*ACC*) downregulated, and eight DEGs including *ACOT*, *mitochondrial enoyl*-[*acyl-carrier protein*] *reductase* (*MECR*), *FabF*, *FabD*, *long-chain fatty acid CoA ligase* (*ACSL*), *carnitine O-palmitoyltransferase 1* (*CPT1*), *carnitine O-palmitoyltransferase 2* (*CPT2*) and *enoyl-CoA hydratase* (*echA*) upregulated under low-temperature stress. RNA-seq and RT-qPCR showed a consistent trend, which indicated that the results of the transcriptome analyses were reliable.

## 4. Discussion

For ectothermic animals, water temperature is one of the important environmental factors. Low-temperature stress can affect amino acid, glycerophospholipid, and nuclei acid metabolism [33], as well as energy metabolism [34]. However, before this study, the potential molecular mechanisms of GFP under low-temperature stress were not fully clear. In the present study, we analyzed changes in genes and metabolites under low-temperature stress for adult GFP. The results of integrated transcriptomic and metabolomic analyses showed that low temperature affected fatty acid (FA) and energy metabolism, which coincided with the result of Hu et al. [35]. Therefore, it is worthwhile to explore the potential mechanism of GFP under low-temperature stress by combining transcriptomic and metabolomic analyses.

### 4.1. Alterations of Lipid Metabolism Associated with Low-Temperature Stress

Lipids, including fatty acyls, glycerolipids, glycerophospholipids, sphingolipids, sterol lipids, prenol lipids, saccharolipids, and polyketides [36], are essential nutrients [37], and energy supplying biological fuels [38]. FA is a vital lipid in animals, which was synthesized by ACC, FAS, and Fatty acid synthase II (Fab). As a key limiting enzyme of FA synthesis, ACC catalyzes acetyl-CoA to malonyl-CoA [39]. Then FAS, which has a vital role in FA synthesis, catalyzes acetyl-CoA and malonyl-CoA into long-chain FA, and its content and activity determine the capacity for FA synthesis [40]. In this study, the expression of both *ACC* and *FAS* genes decreased in LT vs. Con, which was similar to the results in *Cherax quadricarinatus* [41] and GFP [4] under low-temperature stress, indicating that the synthesis of FA was reduced. *Fab* (*Fatty acid synthase II*) including *FabD* and *FabF* besides *MECR* upregulated in LT vs. Con, which catalyzes FA synthesis in mitochondria, and this result suggested that FA was mainly synthesized in mitochondria to cope with low-temperature stress.

In addition, the proportion of unsaturated fatty acid (UFA) is to cell membrane fluidity [42], changing the degree of FA unsaturation was the key physiological adaptation under low-temperature stress [43,44]. The process of integrating polyunsaturated FA into phospholipids to maintain membrane fluidity and metabolic rate at low temperatures is called homeoviscous adaptation [45,46]. In this study, the expression of the gene *HSD17B12* under low-temperature stress increased compared with the control group, and both *ACOT* and *ELOVL* were also upregulated in LT vs. Con, which suggested that biosynthesis of UFA were up-regulated in response to low-temperature stress. Our finding was similar to the results in *M. japonicus* [15], *Takifugu fasciatus* [12], and *Sparus aurata* [47], which further confirmed the theory of homeoviscous adaptation.

For metabolites, FA, such as ALA, LA, icosadienoic acid, and dodecanoic acid, decreased in LS vs. Con, while stearic acid was upregulated. In LT vs. Con, dodecanoic acid was downregulated and traumatic acid was upregulated. ALA and LA are vital constituents of cell membranes and essential for fish [48]. The present results were consistent with the downregulation of ALA under low-temperature stress in *Hemibagrus wyckioides* [14]. Dodecanoic acid can coalesce within the cytoplasm, then change the permeability of the membrane and even induce membrane rupture [49]. Decreased expression of dodecanoic acid might protect the cell membrane. 5-Oxo-ETE generated by lipid peroxidation and enzymatic reaction [50,51], were upregulated in LT vs. LS while no significant changes were noted in both LT vs. Con and LS vs. Con.

Therefore, the current results suggested that low-temperature stress changed lipid metabolism, such as glycerolipid metabolism, fatty acid biosynthesis, and biosynthesis of unsaturated fatty acids. DEGs were upregulated while metabolites had no significant change or were even downregulated, which might have resulted from complex post-transcriptional regulatory mechanisms [52].

### 4.2. Low-Temperature Stress-Induced Energy Metabolism of GFP

In addition to FA metabolism and protein metabolism, energy metabolism is another key metabolism pathway under low temperatures that affects the stress response of aquatic animals [53]. As the basis of growth, the regulation of energy metabolism under low-temperature stress is essential. In this study, DEGs and DMs were enriched in energy metabolism pathways, including glycolysis/gluconeogenesis, TCA cycle, and oxidative phosphorylation.

Playing a vital role in energy metabolism, oxidative phosphorylation can provide about 80% of the total ATP [54]. Environmental stress in marine organisms has a great impact on the amount of energy produced by oxidative phosphorylation [55]. Lipids can provide abundant energy through oxidative phosphorylation [56,57]. FA oxidation is considered an important source of energy production in animals. CPT and ACSL play a vital role in FA oxidation. As a speed-limiting enzyme in FA degradation and mitochondrial β-oxidation, CPT transfers acyl-groups into mitochondria with the acyl-carnitine translocator [58,59]. Long-chain fatty acyl-CoAs are catalyzed from long-chain fatty acids by ACSL [60]. In the previous study, the genes *CPT*, *ACSL*, and *echA* were upregulated under low-temperature stress. Studies noted that *CPT* was upregulated under low-temperature stress in *T. fasiatus* [38] and *ASCL* was upregulated to enhance the hypoxia-tolerant capacity in *Oncorhynchus mykiss* [61]. These results showed that increased expression of *CPT* and *ACSL* might enhance FA β-oxidation for energy under low-temperature stress.

The expression of glycolysis and TCA cycle genes changed under low-temperature stress [62] and hypoxia stress [63]. Glycolysis/gluconeogenesis and the TCA cycle are linked by PCK. First, pyruvate converts to oxaloacetate through the TCA cycle, and then PCK catalyzes oxaloacetate to phosphoenolpyruvate, which can generate glucose in gluconeogenesis [64,65]. Moreover, the TCA cycle is the final metabolism pathway of three nutrients involving carbohydrates, lipids, and amino acids, and glycolysis can produce ATP through the oxidation of hexoses and produce precursors for anabolism, which is the main pathway of anaerobic metabolism [66,67]. As a marker enzyme of anaerobic metabolism, the activity of LDH can represent the level of anaerobic metabolism to a certain extent [68]. In contrast, the aerobic metabolism level can be reflected by the activity of SDH [69], which catalyzes succinic acid to fumarate and produces ATP [70]. The content of succinic acid decreased under low-temperature stress as shown in metabolism analyses, which consisted of the upregulation of SDH in transcriptomic analyses. The increasing expression of *PCK*, *LDH*, and *SDH* in this study may enhance the gluconeogenesis pathway, similar to the results in *Litopenaeus vannamei* [71].

In general, the present results indicated that GFP might provide energy through gluconeogenesis, fatty acid degradation, and the TCA cycle under low-temperature stress.

## 5. Conclusions

The present study showed that DEGs and DMs are mainly enriched in lipid and energy metabolism, such as fatty acid biosynthesis, biosynthesis of unsaturated fatty acid, fatty acid degradation, glycolysis/gluconeogenesis, oxidative phosphorylation, and TCA cycle. Especially, FA plays an important role in coping with low-temperature stress, not only in maintaining the stability and fluidity of cell membranes but also in providing energy. These findings can help us better understand the potential molecular mechanism of low-temperature tolerance in GFP. However, the molecular mechanism of low-temperature tolerance in aquatic organisms like prawns still needs further investigation in the future, such as RNA interference and gene overexpression.

## Figures and Tables

**Figure 1 animals-13-01605-f001:**
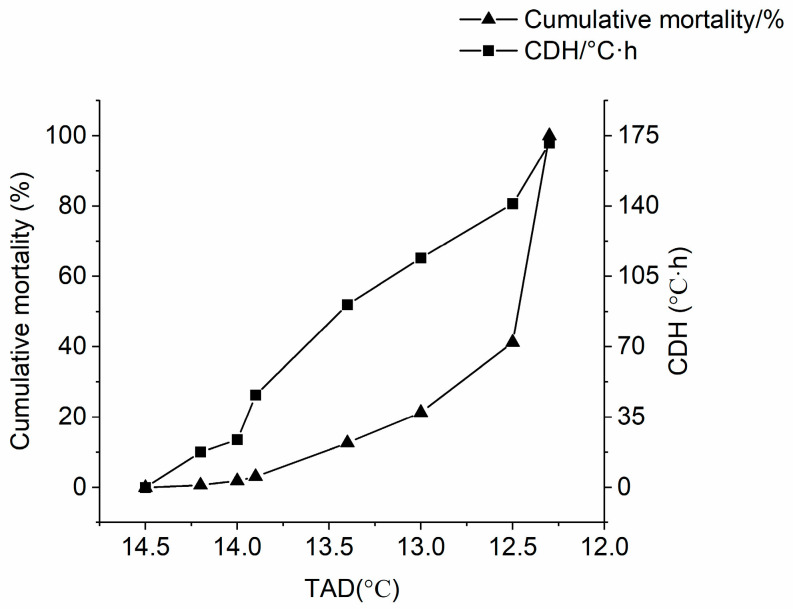
Cumulative mortality and CDH of *M. rosenbergii* under low-temperature stress. TAD represents the temperature at death, CDH represents the cooling degree hours.

**Figure 2 animals-13-01605-f002:**
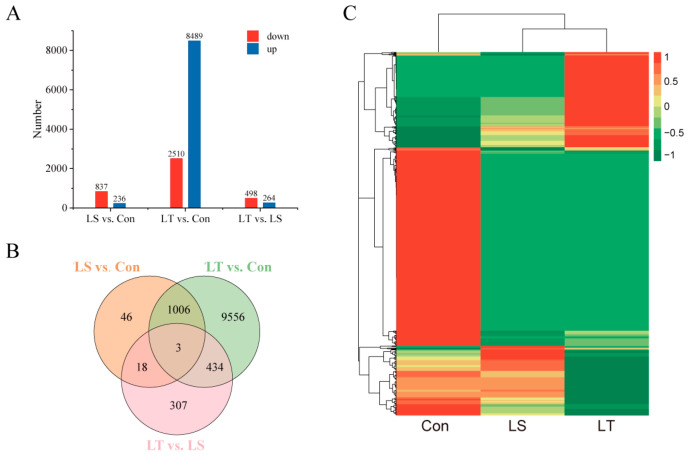
DEGs information in the transcriptome of *M. rosenbergii* under low-temperature stress. (**A**) Number of DEGs; (**B**) DEGs Venn diagram; (**C**) Overlapping DEGs heatmap. LS represents low-temperature sensitive group, LT represents low-temperature tolerant group, Con represents control group.

**Figure 3 animals-13-01605-f003:**
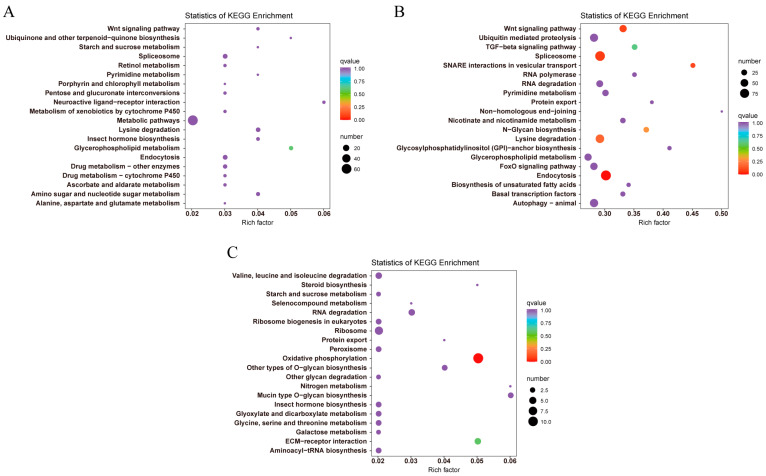
KEGG enrichment TOP20 pathways of *M. rosenbergii* transcriptome under low-temperature stress. (**A**) LS vs. Con; (**B**) LT vs. Con; (**C**) LT vs. LS. LS represents low-temperature sensitive group, LT represents low-temperature tolerant group, Con represents control group.

**Figure 4 animals-13-01605-f004:**
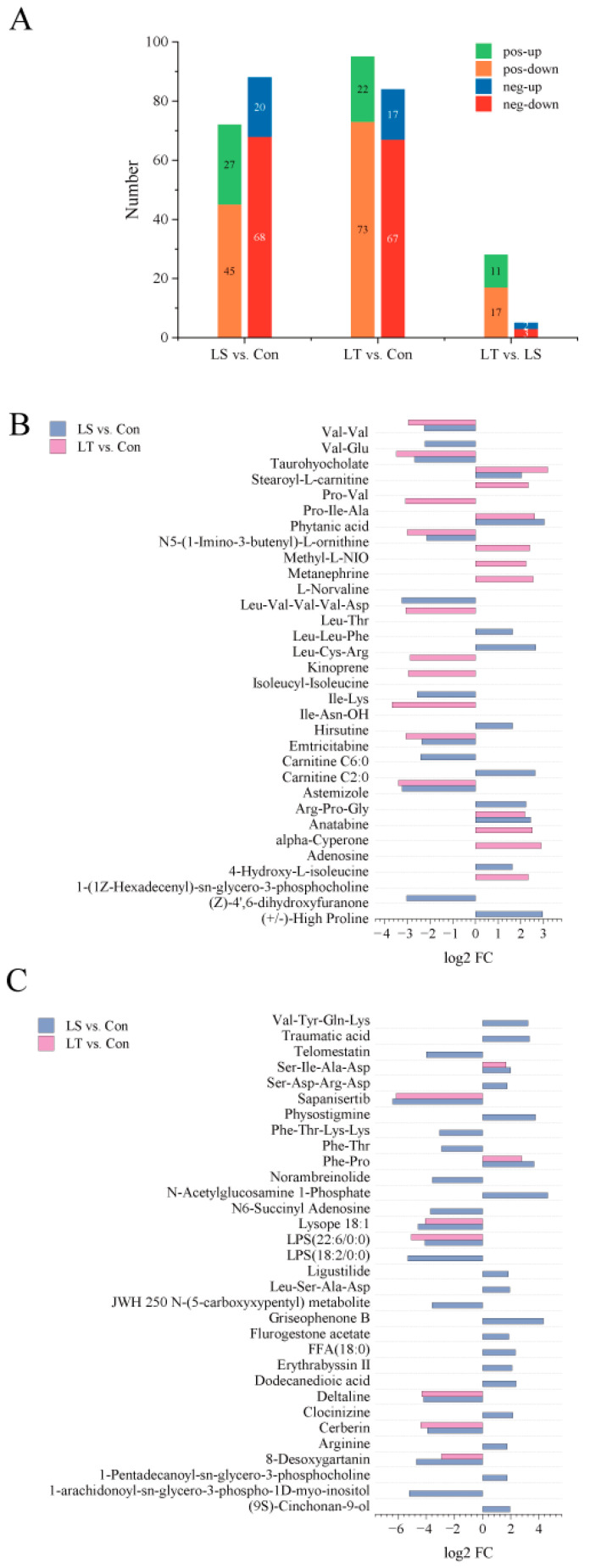
Information of different metabolites (DMs) in the metabolome of *M. rosenbergii* under low-temperature stress. (**A**) Number of DMs; (**B**) TOP20 DMs of LS vs. Con and LT vs. Con in positive ion mode; (**C**) TOP20 DMs of LS vs. Con and LT vs. Con in negative ion mode. LS represents low-temperature sensitive group, LT represents low-temperature tolerant group, Con represents control group.

**Figure 5 animals-13-01605-f005:**
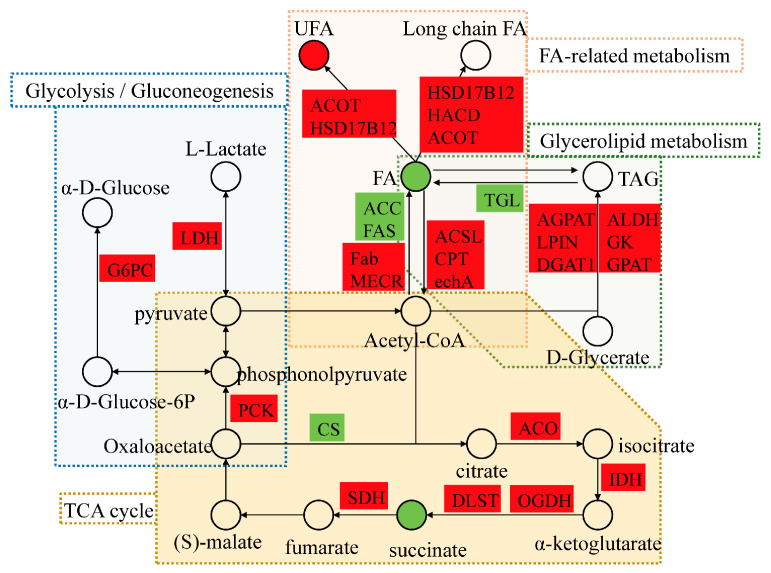
Network diagram of the potential regulatory mechanism of *M. rosenbergii* in response to low-temperature stress. The square represents genes, the circle represents metabolites, the red represents up-regulation, the green represents down-regulation, and the dashed boxes represent the pathways.

**Figure 6 animals-13-01605-f006:**
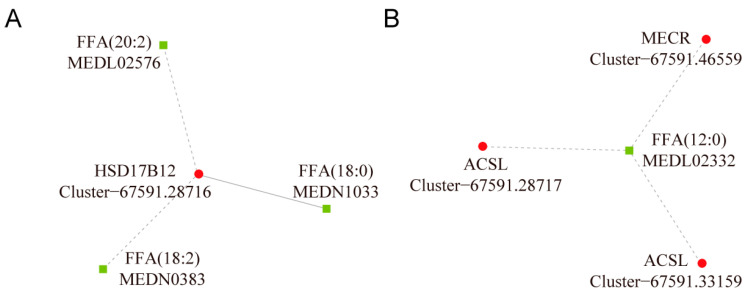
Interaction network of fatty acid metabolism. (**A**) Interaction network diagram of biosynthesis unsaturated fatty acids of LS vs. Con; (**B**) interaction network of fatty acid biosynthesis of LT vs. Con. The square represents metabolites, the circle represents genes, red represents upregulation, green represents downregulation, dashed lines represent negative correlation, and solid lines represent the positive correlation.

**Table 1 animals-13-01605-t001:** Basic information on traits of *M. rosenbergii*.

Trait	Minimum	Maximum	Mean ± SD
TAD/°C	12.30	14.20	12.54 ± 0.41
CDH/°C·h	0.00	171.28	127.39 ± 38.28
BW/g	21.62	88.05	47.73 ± 12.73
BL/mm	70.20	414.87	123.60 ± 16.90

Note: the number of measurements we obtained was 577. TAD represents the temperature at death, CDH represents the cooling degree hours, BW represents body weight and BL represents body length.

**Table 2 animals-13-01605-t002:** The number of DEGs in lipid and energy metabolism pathways of three comparison groups.

Pathway	ko ID	The Number of DEGs		
		LS vs. Con	LT vs. Con	LT vs. LS
Fatty acid biosynthesis	ko00061	0	8	0
Fatty acid elongation	ko00062	1	11	0
Fatty acid degradation	ko00071	0	18	1
Glycerolipid metabolism	ko00561	3	15	0
Glycerophospholipid metabolism	ko00564	8	43	1
Pyruvate metabolism	ko00620	1	16	0
Biosynthesis of unsaturated fatty acids	ko01040	1	13	0
Fatty acid metabolism	ko01212	1	24	1
Glycolysis/Gluconeogenesis	ko00010	3	18	0
Citrate cycle (TCA cycle)	ko00020	1	9	0
Pentose phosphate pathway	ko00030	2	10	0
Oxidative phosphorylation	ko00190	3	50	11

Note: LS represents low-temperature sensitive group, LT represents low-temperature tolerant group, Con represents control group.

**Table 3 animals-13-01605-t003:** The number of DMs in lipid and energy metabolism pathways of three comparison groups. Three comparison groups included LS vs. Con, LT vs. Con, and LT vs. LS.

Pathway	ko ID	The Number of DMs		
		LS vs. Con	LT vs. Con	LT vs. LS
Fatty acid biosynthesis	ko00061	3	1	0
Glycerophospholipid metabolism	ko00564	0	0	1
Arachidonic acid metabolism	ko00590	1	0	1
Linoleic acid metabolism	ko00591	1	0	0
alpha-Linolenic acid metabolism	ko00592	2	1	0
Pyruvate metabolism	ko00620	3	1	0
Butanoate metabolism	ko00650	2	2	0
Biosynthesis of unsaturated fatty acids	ko01040	4	0	1
Regulation of lipolysis in adipocytes	ko04923	1	1	0
Oxidative phosphorylation	ko00190	1	1	0
Citrate cycle (TCA cycle)	ko00020	2	1	0

Note: LS represents low-temperature sensitive group, LT represents low-temperature tolerant group, Con represents control group.

## Data Availability

All supporting data are included within the main article and its Appendix A.

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
