# Peer review of "Integrated Transcriptomic and Metabolomic Analyses Reveal Low-Temperature Tolerance Mechanism in Giant Freshwater Prawn Macrobrachium rosenbergii"

_animals, 2023, doi:10.3390/ani13101605_

Round 1
Reviewer 1 Report
The paper is important in that it explains the molecular mechanisms behind the response of FW prawns to stressors such as low temperature as this environmental factor can adversely affect survivability of prawns in culture. Comments have been made directly on the paper for appropriate action by the authors.

Reviewer 2 Report
The authors address low temperature tolerance in a warm-temperature organism. only few literature is available for this study but much on high temperature tolerance as a result of climate. I consider that the topic is original or relevant in the field.
The methodology is okay as is, but more can be designed in another study.
Line 68 to 71 is not relevant to the preceding contents.
Add a few data relating to transcriptomic and metabolomic analysis, since your topic emphasized the integration of both omics and give a little introduction to the relevance and importance of integrating both omics in revealing tolerance mechanism.
Enlarge Figure 4, the texts are too small and blur.
The conclusion is consistent with the evidence and arguments presented, and it addresses the main question posed.
The references are appropriate.
Reviewer 3 Report
Dear Editor,
The manuscript entitled “Integrated transcriptomic and metabolomic analyses reveal low-temperature tolerance mechanism in giant freshwater prawn Macrobrachium rosenbergii” by Haihui Tu et al. presents a study focused to assessment of the molecular mechanism of responding to the low-temperature stress in adult giant freshwater prawn through transcriptomic and metabolomic analyses, with experimentation in survival temperature ranges from 18 ℃ to 34 ℃. The authors found that differentially expressed genes and metabolites were both enriched in lipid and energy metabolism pathways. In fact, the levels of unsaturated fatty acids were decreased at the beginning of the cooling period while in the late cooling stage, the genes related to fatty acid synthesis and degradation were upregulated to cope with low-temperature stress.
Τhe manuscripts’ objects are interesting, it well written in a comprehensive way and the findings are interesting and justified. Therefore, the manuscript could be accepted for publication after some minor revisions:
1. In section 2.1, line 83: the authors state that they used 22 prawn families, but what is the differentiation point between the families? Their low-temperature tolerance or other trait?
2. In section 2.1, lines 92-93: the authors report that “The first and last 5 % of mortalities were defined as the low-temperature sensitive group (LS) and the low-temperature tolerant group (LT), respectively”. What is the temperature limits for these groups?
3. In section 2.1, line 94: what is the control group characteristics, i.e. temperature, mortality, etc.
4. In section 2.5, line 152: the gene names should be added in table S1, not only the gene symbols.
5. Please explain the meaning of the cooling degree hours use, I cannot understand why you use it.
6. In table 1: please clarify what you mean by number of record. you mean the measurement you obtained? In that case this can be added as a footnote of the table, it doesn’t have to be a separate column.
7. Throughout the manuscript the authors use 2 different symbols for the experimental groups: LT & LS or CT &CS. Please choose one to use in the text and the figure-table legends.
8. Figure legends need to be more descriptive, they need more details.
9. In page 6, line 224: you mean both experimental groups? Please clarify.
10. In figure 4A, please change colors to differentiate positive and negative modes.
